# SDiD: Shared Diffusion Prior for
# Efficient Distribute Stereo Image Compression

**Yichong Xia** [1 2]  **Yimin Zhou** [1]  **Zongyu Li** [3]  **Shiyu Qin** [1]  **Mingyao Hong** [2]  **Bin Chen** [4]  **Haoqian Wang** [1]

## Abstract

Stereo vision is widely utilized in automotive imagery and 3D reconstruction, creating a demand for compressing stereo images. Existing methods for stereo image compression often employ VAE-like architectures based on distortion optimization, leading to subpar perceptual quality at low bitrates. While generative compression excels in high perceptual fidelity at low bitrates, it struggles to maintain consistency across viewpoints, making decoded images less useful for critical downstream tasks. To address this, we introduce SDiD, a distributed stereo image compression architecture based on shared pre-trained diffusion priors. We employ a diffusion prior alignment module to efficiently obtain the mainview-prior from the foundation diffusion, and utilize a prior transformation structure to enable the auxiliary view to achieve reliable and fast perceptual enhancement while maintaining consistency. Through extensive experiments, we demonstrate that SDiD outperforms existing methods in perceptual quality across multiple datasets. Even at extremely low bitrates, SDiD can accurately recover depth information between decoded images. On the InStereo2K dataset, SDiD requires only one-third of the bits compared to the state-of-the-art baseline (0.02 bpp vs. 0.06 bpp) to reconstruct image pairs with similar depth information.

## 1. Introduction

With the rapid advancement of stereoscopic imaging technologies and the widespread adoption of binocular imaging

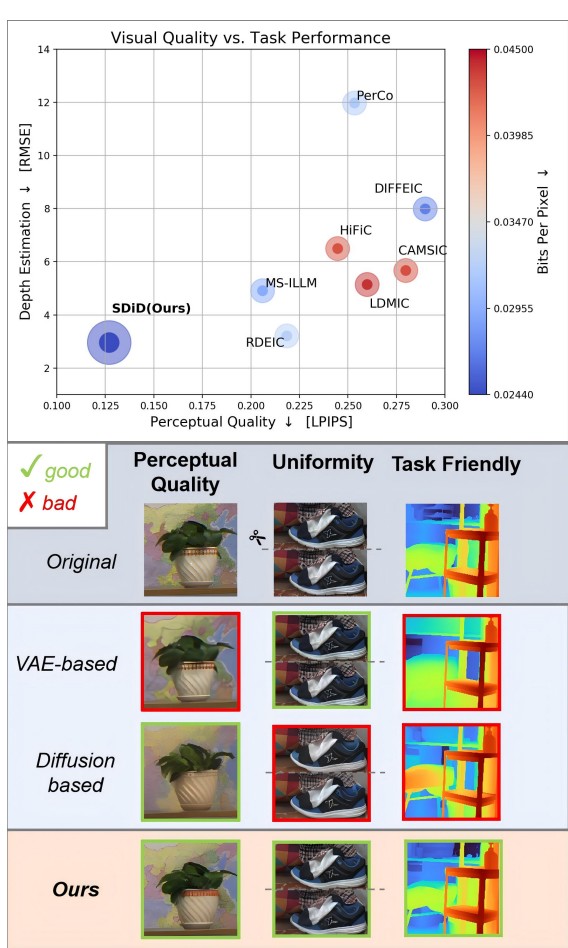

*Figure 1.* Qualitative and quantitative comparison between SDiD and various baselines. Under the premise of the lowest bitrate, our scheme achieves significantly optimal performance in the downstream task (depth estimation) and the best perceptual quality. The visualization results show that SDiD combines the consistency of VAE-based methods and the realism of generative methods, and the decoded images also perform well in downstream tasks.

[1]Tsinghua Shenzhen International Graduate School, Tsinghua University [2]Peng Cheng Laboratory [3]Harbin Institute of Technology, Weihai [4]Harbin Institute of Technology, Shenzhen. Correspondence to: Mingyao Hong <hongmy@pcl.ac.cn>, Bin Chen <chenbin2021@hit.edu.cn>.

*Proceedings of the $43^{rd}$ International Conference on Machine Learning*, Seoul, South Korea. PMLR 306, 2026. Copyright 2026 by the author(s).

devices, stereo images have gained extensive utilization in critical domains such as autonomous driving, augmented reality, and robotic navigation. In this context, the efficient handling and transmission of vast amounts of stereoscopic image data have emerged as a pressing and pivotal issue in need of urgent resolution. For instance, in the realm

of autonomous driving, vehicles equipped with binocular cameras can generate approximately 1GB of data per second, presenting significant challenges to storage and transmission systems. Therefore, the development of efficient Stereoscopic Image Compression (SIC) technology is of paramount importance.

Unlike single-image compression, the main advantage of SIC lies in that it can not only reduce the redundancy within a single image (such as spatial redundancy and texture redundancy) through traditional image compression methods, but also make full use of the correlations between images from different viewpoints in a binocular system (such as disparity correlation and structural correlation) to improve coding efficiency further. At present, most deep learning-based SIC methods still follow the design concept of existing multi-view coding standards, such as the Multi-View High Efficiency Video Coding (MV-HEVC) standard based on HEVC (Tech et al., 2015). These methods usually adopt a joint coding structure to compress multi-view images. The recent advancements in stereoscopic image compression have greatly benefited from the progress in deep single-image compression algorithms (Ballé et al., 2017; 2018) and stereo matching techniques. Some approaches utilize traditional unidirectional encoding methods, such as (Liu et al., 2019a), (Deng et al., 2021a), (Wödlinger et al., 2022), (Qin et al., 2025) and (Zhang et al., 2025). These methods strictly adhere to a sequential encoding process, transmitting the latent representations of auxiliary views as context to the encoding branch of the main view and employing either disparity estimation or depth homography estimation to eliminate redundancies.

However, the scenario of SIC is often asymmetric. This implies that the encoding end needs to be relatively lightweight for easy deployment on terminal devices, while the decoder deployed in the cloud can be more complex to perform resolution for various downstream tasks. Joint encoding SIC schemes can lead to an unacceptable encoding burden. Distributed Source Coding (DSC) is a class of excellent solutions: according to the theory of DSC (Slepian & Wolf, 1973; Wyner & Ziv, 1976), independently encoding correlated data sources and utilizing side information at the decoder can achieve the same compression rate as joint encoding. To achieve this asymmetric structure, LDMIC uses cross-attention to capture overlapping information between viewpoints, while MSFDPM (Huang et al., 2023) and FCA-Net (Xia et al., 2025a) employ a patch-matching method to capture contextual information between images. These methods have shown satisfactory performance at high bitrates, but struggle to ensure visual quality and semantic consistency in decoded images at low bitrates. Compression schemes based on fine-tuning foundational diffusion models (Careil et al., 2024; Xia et al., 2025b; Li et al., 2024; Ke et al., 2025; Li et al., 2025b; Xia et al., 2026) have

demonstrated surprising perceptual fidelity in low and extremely low bitrate scenarios, with the latency introduced by denoising primarily concentrated in the decoding process, aligning well with the application scenarios of stereo images. However, many studies indicate that diffusion struggles to guarantee temporal consistency within the same scene. As shown in Figure 1, compression schemes based on pre-trained diffusion face challenges in ensuring object consistency between viewpoints, directly impacting the difficulty of using decoded images for downstream tasks such as depth estimation.

The pre-trained diffusion architecture is highly suited for application in DSC-based SIC; our work aims to address the dual shortcomings of such approaches. (1) Loss of precision in downstream tasks due to inconsistencies between decoded image viewpoints. (2) The exorbitant denoising costs incurred by denoising networks. From this standpoint, we introduce the **S**hared **Di**ffusion Prior for Efficient **D**istributed Stereo Image Compression (SDiD). SDiD independently encodes images from stereoscopic viewpoints and employs a controllable denoising network to recover information from the main viewpoint. Subsequently, the shareable prior representation enhances the auxiliary view through a transfer module. Specifically, we reflect on the training paradigm of $\epsilon$-prediction in foundational diffusion models, utilizing a diffusion prior alignment module to actively align the denoising network's output with compressed representations, achieving rapid two-step sampling while optimizing bitrate allocation. Furthermore, we introduce Look-back Attention to effectively capture the inter-view context. Experimental results demonstrate that our approach significantly surpasses existing compression baselines in both perceptual quality and metrics for downstream depth estimation tasks. Perceptual fidelity enhancements range from 20% to 120% compared to generative baselines, and improvements in depth estimation performance of decoded images range from 30% to 90% compared to traditional SIC methods. The mechanisms of accelerated sampling and shared priors enable us to achieve a 50% acceleration in decoding compared to other few-step diffusion models and a 15-fold acceleration compared to the typical 50-step sampling paradigm.

Our contributions can be summarized as follows:

- We introduced Diffusion Prior Alignment (DPA) to rectify and enhance the pre-trained diffusion's prior, aligning it with the compressed latent space. By utilizing Adaptive Noise Estimation (ANE) to gauge the noise level of the input compressed representation, we swiftly generated a shareable main viewpoint prior.

- We proposed Diffusion Prior Transfer (DPT) incorporates Look Back Attention (LBA) to explicitly cap-

ture correlations between inter-view representations. This approach enables the precise transformation of the main-viewpoint-prior to the auxiliary viewpoint, facilitating the compressor to acquire more compact latent representations during implicit training.

- Building on these insights, we presented **S**hared **Di**ffusion Prior for Efficient **D**istributed Stereo Image Compression (SDiD). Extensive experiments confirm that SDiD consistently achieves optimal results on InStereo2K and Cityscapes datasets. Compared to VAE-based DSC schemes, SDID demonstrates a 255% increase in perceptual metrics and a 48% increase in performance in downstream depth estimation tasks.

## 2. Related work

### 2.1. Neural image compression

Learnable image compression methods (Ballé et al., 2017; 2018; Minnen et al., 2018; He et al., 2022; Cheng et al., 2020; Qin et al., 2024; 2026; Zhang et al., 2026) involve extracting image representations using an encoder, estimating the probability distribution of these representations using an entropy model, and performing entropy coding. According to Shannon's information theory, the training paradigm can be unified into the following form:

$$\mathcal{L} = R_{\boldsymbol{x}} + \lambda D(\boldsymbol{x}, \mathcal{C}(\boldsymbol{x})) \tag{1}$$

Here, $\mathcal{C}(\cdot)$ represents the compressor, $R$ denotes the bitrate of the image $\boldsymbol{x}$. $D(\cdot)$ typically chooses pixel-level distortions such as MSE or MS-SSIM (Wang et al., 2003). However, distortion-based losses often lead to changes in the distribution between reconstructed and natural images (Blau & Michaeli, 2019), prompting research into enhancing perceptual quality. HiFiC (Mentzer et al., 2020) explores the integration of GANs (Goodfellow et al., 2020) with compression. MS-ILLM (Muckley et al., 2023) enhances reconstruction quality through an improved discriminator design. Constrained by GAN structures and a single perceptual loss, these methods still have significant room for improvement in statistical fidelity.

In comparison, diffusion models, such as DDPM(Ho et al., 2020), demonstrate superior generative quality through a progressive denoising process, with the statistical properties of their reconstructed images aligning more closely with natural image distributions. The CDC (Yang & Mandt, 2024) research team pioneered the application of the DDPM framework in the field of image compression, achieving breakthroughs; however, this method requires complete training of diffusion model components, leading to challenges such as high computational costs and limited generalization due to training data constraints. More recently, some image compression works based on Latent Diffusion Models (LDM)

(Rombach et al., 2022) have introduced new perspectives. (Lei et al., 2023) attempts to control pre-trained LDM by encoding sketches and text semantics, sampling image recovery within the diffusion framework during the decoding process. Given the challenges in jointly optimizing semantic embeddings and compression objectives, they heavily invest in iterative semantic embedding and alignment. (Careil et al., 2024) employs a fully trained conditional LDM and encodes conditional images using a trainable codebook. On the one hand, this necessitates meticulous model optimization on datasets comprising millions of images. On the other hand, the training of the compressor and diffusion model is independent, limiting the compressor's learning capacity. (Li et al., 2024; 2025b) utilize pre-trained foundational LDMs, yet their joint training approach makes it difficult for the compressor to leverage diffusion priors for learning compact representations. (Ke et al., 2025) harnesses Multimodal Large Language Models (MLLM) to obtain high-quality textual semantics, achieving significant performance gains at ultra-low bitrates, albeit with extremely high encoding and decoding latency.

In estimating these compression tasks based on foundational diffusion, the compressor replaces the encoded image with features extracted by encoding pre-trained analytical encoders. At this point, the training objective is replaced with:

$$\mathcal{L} = R_{\mathcal{E}(\boldsymbol{x})} + \lambda D(\mathcal{E}(\boldsymbol{x}), \mathcal{C}(\mathcal{E}(\boldsymbol{x}))) \tag{2}$$

In this context, $\mathcal{E}(\cdot)$ represents the pre-trained analytic encoder. Although this method can streamline the compressor, the inconsistency between distortion in the feature domain and distortion in the image domain may lead the optimization to the error of the optimal rate-distortion trade-off solution.

### 2.2. Diffusion models

Diffusion models, or score-based generative models, are a class of generative models that gradually inject Gaussian noise into data and then generate samples from the noise through a reverse denoising process.

Latent Diffusion Models (LDMs) (Rombach et al., 2022) address the inefficiency of pixel-space-based generative models by operating in a lower-dimensional latent space. This design reduces the computational complexity of the diffusion process while preserving the ability to model high-fidelity image distributions, thereby enabling more efficient and scalable generation of visual content. In the forward diffusion process of LDMs, Gaussian noise is incrementally injected into the clean latent feature $\boldsymbol{z}_0$ over a predefined number of timesteps $T$. The intensity of the noise added at each timestep $t \in \{1, 2, \ldots, T\}$ is regulated by a precomputed noise schedule $\{\beta_t\}_{t=1}^{T}$, where $\beta_t \in (0, 1)$ and typically satisfies $\beta_1 < \beta_2 < \cdots < \beta_T$ to ensure gradual

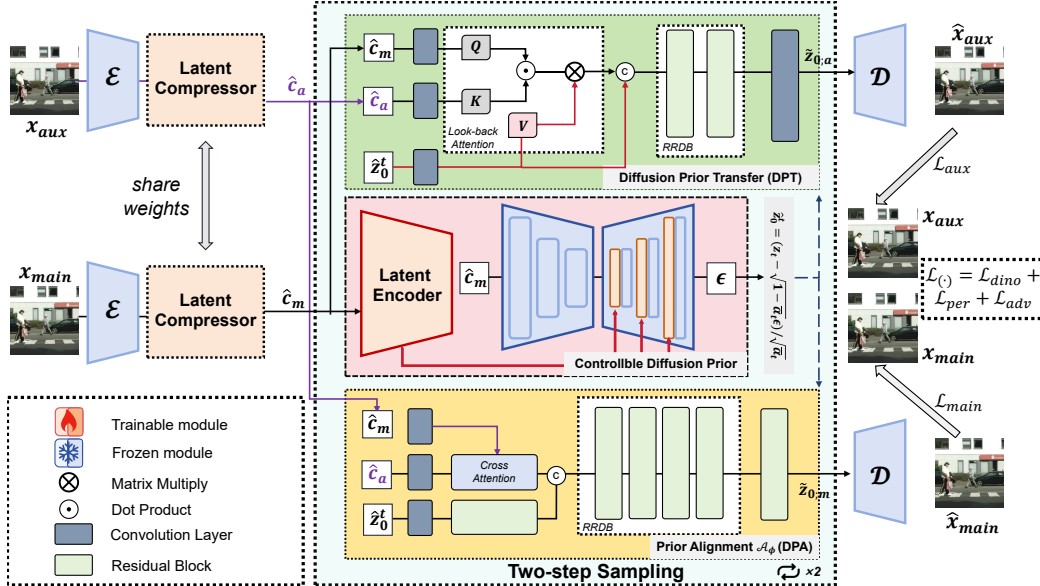

*Figure 2.* Illustration of the proposed SDiD. RRDB signifies Residual in Residual Dense Block (Wang et al., 2018), Latent Encoder is derived from (Li et al., 2025a). SDiD adopts a distributed architecture. The encoding end employs independent latent compressors with shared parameters, while the decoding end exchanges inter-view contextual information and shares the prior of the foundation-diffusion model.

noise accumulation. This forward process is mathematically formulated as follows:

$$z_t = \sqrt{\bar{\alpha}_t} z_0 + \sqrt{1 - \bar{\alpha}_t} \epsilon, \quad t \in \{1, 2, \ldots, T\}, \quad (3)$$

where $\epsilon \sim \mathcal{N}(0, \mathbf{I})$ is standard Gaussian noise. Here, $\alpha_t = 1 - \beta_t$ and $\bar{\alpha}_t = \prod_{i=1}^{t} \alpha_i$. As $t$ increases, the corrupted $z_t$ progressively approximates a Gaussian distribution. The reverse process of modeling LDMs is equivalent to train a noise prediction network $\epsilon_\theta$ with diffusion loss:

$$\mathcal{L}_{\text{diff}} = \mathbb{E}_{z_0, t, \epsilon \sim \mathcal{N}(\mathbf{0}, \mathbf{I})} \|\epsilon - \epsilon_\theta(z_t, c, t)\|_2^2 \quad (4)$$

During inference, LDMs predict noise using the pre-trained denoising network $\epsilon_\theta(z_t, c, t)$ with text condition $c$, generating latent $z_{t-1}$ to sequentially obtain the final latent $z_0$.

**2.3. Stereo image compression**

Recently, learning-based paradigms for stereo image compression have evolved rapidly. Unidirectional coding (Liu et al., 2019b; Deng et al., 2021b; Wödlinger et al., 2022; Zhai et al., 2022) encodes one view independently and reconstructs the other through disparity- or featurebased prediction, whereas bidirectional coding (Lei et al., 2022; Wödlinger et al., 2024; Liu et al., 2024) jointly compresses both views via cross-view feature interaction, achieving better rate–distortion trade-offs and balanced quality. Moreover, several recent works focus on entropy model optimization rather than network design. For instance, CAMSIC

(Zhang et al., 2025) employs a content-aware, decoder-free transformer entropy model, forming a neat yet powerful framework. The latest study, SICM (Jin et al., 2025), further extends stereo compression toward machine vision through a feature-oriented stereo network (MVSFC-Net) that encodes compact stereo representations for 3D perception tasks, achieving notable bitrate savings and task-oriented performance gains.

## 3. Methodology

### 3.1. Framework overview

Our proposed SDiD framework, as illustrated in Figure 2, operates as follows: At the encoding stage, features from both the main and the auxiliary viewpoints are initially extracted through $\mathcal{E}(\cdot)$ and then compressed using an independent feature compressor with shared parameters, producing compressed representations transmittable with loss $\hat{c}_{main}, \hat{c}_{aux}$ (also denoted as $\hat{c}_m, \hat{c}_a$). At the decoding stage, the recovered $\hat{c}_m, \hat{c}_a$ are used together as conditions to guide the pre-trained denoising network in generating priors for the main viewpoint. Subsequently, the main viewpoint prior is refined and enhanced for the auxiliary viewpoint's representation through a diffusion prior transformation module. Finally, the refined multi-viewpoint representations are decoded by the pre-trained analytical decoder $\mathcal{E}(\cdot)$ to obtain the final images. We utilize the codebook-based representation compressor proposed by (Li et al., 2025b) as our latent compressor, with its entropy model structure derived from

(He et al., 2022).

## 3.2. Diffusion prior alignment (DPA)

We have employed the control module from (Li et al., 2025a) to oversee the denoising network, facilitating the generation of diffusion priors for the main viewpoint.

However, the priors obtained from pre-trained diffusion are misaligned, making it challenging for us to learn the prior distribution during end-to-end training. Typically, we utilize proxy losses to prevent mode collapse. However, this hinders our ability to achieve the optimal bitrate distribution. To address this issue, we have devised Diffusion Prior Alignment (DPA) $\mathcal{A}\phi$, mapping the denoising results at any time point back to the initial point $z_0$:

$$\mathcal{A}_\phi\left(\hat{z}_{0;m}^t, \hat{c}_m, \hat{c}_a, t\right) = \tilde{z}_{0;m}^t \sim z_{0;m} \quad (5)$$

where $z_{0;m} = \mathcal{E}(x_m)$. Here, $\hat{z}_{0;m}^t$ is obtained from the noise output by the denoising network and $\epsilon$-prediction:

$$\left(z_{t;m} - \sqrt{1 - \bar{\alpha}_t}\epsilon_\theta^t\right)/\sqrt{\bar{\alpha}_t} = \hat{z}_{0;m}^t \quad (6)$$

This approach ensures that the compressor can eliminate redundancies overlapping with the prior during training and enables the establishment of a broader range of training objectives to supervise the training process. To enable this module to achieve accelerated sampling, we must ensure the consistency (Song et al., 2023) of $\mathcal{A}_\phi$ during the training process:

$$\mathcal{L}_\mathcal{C}\left(\phi, \phi^-\right) = \\ \mathbb{E}_{z,\hat{c},n}\left[d\left(\mathcal{A}_\phi\left(\epsilon_\theta^{t_{n+k}}, \hat{c}_m, \hat{c}_a, t_{n+k}\right), \mathcal{A}_{\phi^-}\left(\epsilon_\theta^{t_n}, \hat{c}_m, \hat{c}_a, t_n\right)\right)\right] \quad (7)$$

$d(\cdot, \cdot)$ is MSE, with $k = 20$. Unlike consistency distillation, $\mathcal{A}_\phi$ is initialized to zero, necessitating the loss of prediction $z_0$ to stabilize the prediction process:

$$\mathcal{L}_\mathcal{A} = w(t) \cdot \|\mathcal{A}_\phi(\epsilon_\theta^t, \hat{c}_m, \hat{c}_a, t) - z_{0;m}\|_2^2 \quad (8)$$

Here, $w(t) = \frac{1}{2}\left(\frac{\bar{\alpha}t - 1}{1 - \bar{\alpha}t - 1} - \frac{\bar{\alpha}_t}{1 - \bar{\alpha}_t}\right)$.

The predicted value of $z_0$ obtained through $\epsilon$-Prediction is referred to as $\hat{z}_0$. To mix information from both viewpoints during the prior alignment phase, the compressed representation of the auxiliary viewpoint will be concatenated with the compressed representation of the main viewpoint using cross-attention. These, along with the diffusion prior, are fed into the DPA module to obtain the final prediction $\tilde{z}_0$.

## 3.3. Adaptive noise-level estimator (ANE)

In traditional diffusion frameworks, the denoising process begins with sampling from a Gaussian distribution. We have found that starting the denoising process from a Gaussian is

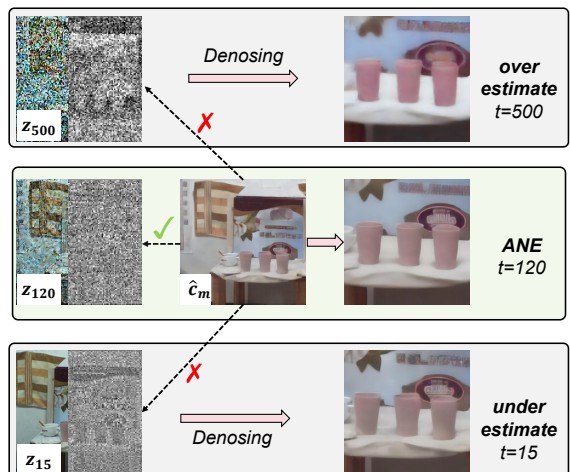

*Figure 3.* (Left) The visualization results of different temporal states during the forward process, along with the differences between these results and the $\hat{c}_m$. It can be observed that ANE accurately estimates the temporal state, and the differences do not contain any image structures. (Right) Denoising results using this temporal state as input; it is evident that either an excessively high or excessively low estimation of the temporal state will lead to poor results (either over-blurring or edge distortion).

not suitable for employing aggressive stride sampling strategies. As noted in (Ke et al., 2025), the states of early time steps have limited capacity to carry original information. While this may reduce the bitrate, it is challenging to ensure reconstruction quality. To achieve efficient two-step sampling, we propose Adaptive Noise Level Estimation (ANE). We use $\hat{c}_m$ directly as the initial input for the denoising network instead of Gaussian noise. To align with the denoising patterns of the pre-trained diffusion network, we need to estimate the noise level contained in $\hat{c}_m$. Specifically, we calculate the signal-to-noise ratio using $\hat{c}_m$ and the lossless representation $z_{0;m}$:

$$\text{SNR}_{\hat{c}} = \|z_{0;m}\|_2^2/\|\hat{c}_m - z_{0;m}\|_2^2 \quad (9)$$

We can calculate the signal-to-noise ratio for each time step state in the DDPM framework using the forward noising formula (Luo, 2022):

$$\text{SNR}_F(t) = \frac{\bar{\alpha}_t}{1 - \bar{\alpha}_t} \quad (10)$$

Then, we estimate the approximate time step $\hat{t}$ corresponding to the noise level of $\hat{c}_m$ by finding the $\text{SNR}_F(t)$ with the smallest difference from $\text{SNR}_{\hat{c}}$.

$$\hat{t}(\hat{c}_m) = \arg\min_t \|\text{SNR}_F(t) - \text{SNR}_{\hat{c}}\|_1 \quad (11)$$

In fact, Equation (11) is equivalent to finding the minimum Wasserstein distance between the compressed representation distribution and the forward diffusion path. Please refer to the appendix for detailed proof.

To address the challenge of searching for approximate time steps, we have subdivided the original 1000 steps into 50 time steps with intervals of 20. During the transmission process, $\hat{t}$ will be encoded and transmitted simultaneously. Our proposed noise estimation method effectively adapts to the denoising patterns of the pre-trained framework, as depicted in Figure 3. ANE accurately estimates the noise level of the denoising object. Deviations in denoising results can occur with either larger or smaller time step estimations. In our ablation experiments, the utilization of ANE resulted in greater performance gains compared to using the original noise schedule.

### 3.4. Diffusion prior transfer (DPT)

In low-bitrate scenarios, ensuring consistent reconstruction of the same object across different viewpoints poses a significant challenge. To address this issue, we have introduced a prior transformation module to facilitate the effective transfer of the main viewpoint prior to the auxiliary view.

The DPT module leverages Look-Back Attention (LBA) and dense residual blocks to enhance the decoding of the auxiliary viewpoint. In this process, we employ a resource-efficient attention mechanism, using $1 \times 1$ convolutions to transform $\hat{c}_a$ and $\hat{z}_0^t$ into query $Q_{aux}$ and value $V_{prior}$ terms, respectively. Notably, in extremely low bitrate scenarios, we have observed a notable disparity between the main viewpoint representation $\hat{z}_0^t$ generated by the denoising network and the compressed latent $\hat{c}_a$, rendering the computation of attention maps between them unreliable. Hence, we opt to utilize $\hat{c}_m$ to generate the key term $K_{main}$ to obtain the final outcome.

$$\text{LBA}(\hat{c}_a, \hat{c}_m, \hat{z}_0^t) = \text{Softmax}\left(\frac{Q_{aux}K_{main}}{\sqrt{d}}\right) \cdot V_{prior} \quad (12)$$

Since $\hat{c}_a$ and $\hat{c}_m$ are encoded by independent encoders with similar compression rates and thus have similar distributions, a relatively stable attention map can be computed. Experiments have shown that LBA exhibits a significant performance improvement compared to standard cross-attention.

### 3.5. Loss function

The training objective is divided into three parts, with the loss in the first part supervising the codebook-based compressor. We use $\odot$ to represent one of the viewpoints in a pair of stereo images.

$$\mathcal{L}_{\text{comp};\odot} = \lambda_1 \|\tilde{z}_{0;\odot} - z_c\|_2^2 + \lambda_2 R_{x_\odot} + \mathcal{L}_{\text{codebook}} \quad (13)$$

The second part is utilized to constrain the control encoder and DPA module, aiming to preserve their self-consistency while maintaining their $z_0$-Prediction functionality.

$$\mathcal{L}_{\text{DPA}} = \lambda_1 \mathcal{L}_{\mathcal{A}} + \mathcal{L}_{\mathcal{C}}\left(\phi, \phi^-\right) \quad (14)$$

The third segment concerns the reconstruction loss. Following the guidelines from (Wu et al., 2024; Lee et al., 2024), we calculate the loss $\mathcal{L}_{dino}$ as the disparity between the reconstructed image and the original image in the DINOv2 (Oquab et al., 2023) feature domain. We also incorporate perceptual loss $\mathcal{L}_{per}$ and optimize the training process using GAN loss $\mathcal{L}_{adv}$.

$$\mathcal{L}_{rec;\odot} = d_1\mathcal{L}_{dino}(x_\odot, \hat{x}_\odot) + d_2\mathcal{L}_{per}(x_\odot, \hat{x}_\odot) + d_3\mathcal{L}_{adv} \quad (15)$$

In conclusion, the training loss for SDiD is:

$$\mathcal{L}_{\text{SDiD}} = \mathcal{L}_{\text{comp};m} + \mathcal{L}_{\text{comp};a} + \mathcal{L}_{\text{DPA}} + \mathcal{L}_{rec;m} + \mathcal{L}_{rec;a} \quad (16)$$

## 4. Experiments

### 4.1. Datasets

Our proposed method has been validated on two high-resolution stereo image datasets, specifically including Cityscape-valid (Cordts et al., 2016) (which focuses on outdoor distant-view scenarios) as well as InStereo2K-test (Bao et al., 2020) (which targets indoor near-view scenarios).

Cityscape-valid consists of 500 stereo image pairs with a resolution of $2048 \times 1024$ pixels. InStereo2K includes 2,010 training stereo image pairs and 50 testing stereo image pairs, with each image having a fixed resolution of $1080 \times 860$ pixels.

### 4.2. Baseline

We shall compare our proposed SDiD against a myriad of representative image compression methods, encompassing VAE-based DSC method LDMIC (Zhang et al.); VAE-based SIC method CAMSIC (Zhang et al., 2025), ECSIC (Wödlinger et al., 2024); GAN-driven compression techniques including HiFiC (Mentzer et al., 2020) and MS-ILLM (Muckley et al., 2023); alongside diffusion-based approaches such as PerCo (Careil et al., 2024), DiffEIC (Li et al., 2024), RDEIC (Li et al., 2025b). Further details can be perused in the appendix.

### 4.3. Metric

Multiple evaluation metrics were employed to fully assess the performance of the model. Similar to other compression tasks, we used bits per pixel (bpp) as a metric to measure the degree of compression. Based on the evaluation types, metrics can be categorized into two classes. *(1) Reference-based perceptual-based metrics*: LPIPS(Zhang et al., 2018) and DISTS(Ding et al., 2020). These metrics can effectively reflect the overall image quality and the reconstruction performance as perceived by human vision. *(2) Downstream-task-based metrics*: To validate the performance of the decoded images in the critical downstream task: depth estimation, we refer to the evaluation metrics in (Wen et al.,

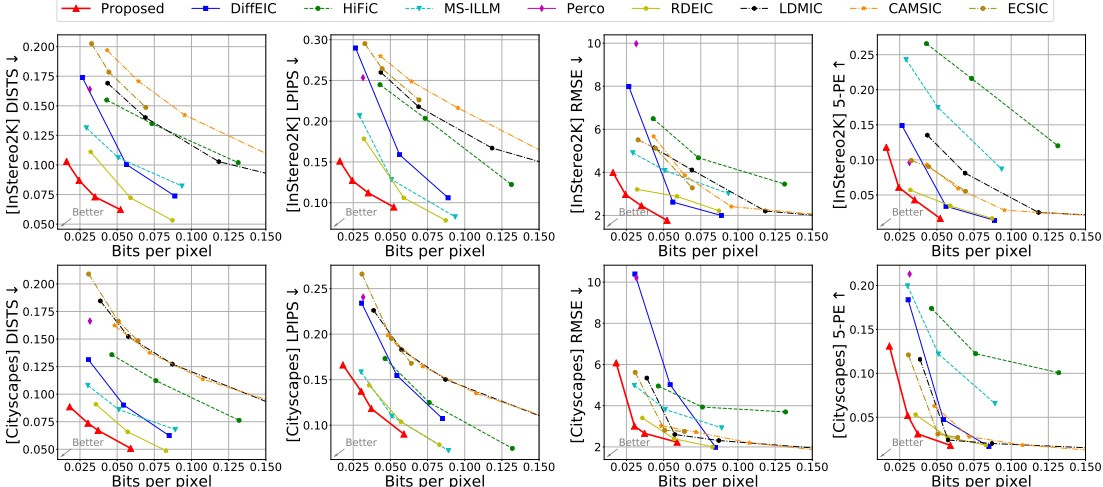

*Figure 4.* Qualitative comparison on the Cityscapes and InStereo2K datasets. Grey arrows indicate the direction of better performance. The indicators within the red box represent perceptual quality, while those within the blue box represent downstream task performance.

2025; Niccoli et al., 2025). Specifically, two key metrics are employed to quantify performance: Root Mean Square Error (RMSE), calculated between the depth estimation maps derived from the decoded images and those from the original images. 5-Pixel Error Percentage (5-PE): Computes the percentage of pixels where the disparity error exceeds 5 pixels. The downstream depth estimator we adopted is FoundationStereo (Wen et al., 2025), which has been verified as a state-of-the-art stereo image depth estimation solution with high generalization.

### 4.4. Experimental results

#### 4.4.1. CODING PERFORMANCE

Figure 4 illustrates the RD curves for SDiD in comparison with all baseline models, while the Table 2 presents the BD-Rate for each baseline. BD-Rate is calculated with respect to SDiD, where a higher value indicates a greater performance deficit relative to SDiD. Notably, in both datasets, SDiD consistently demonstrates superior performance in both human and machine vision. In contrast to the VAE-structured SIC methods, CAMSIC and LDMIC, the baseline models based on generative models exhibit markedly better perceptual quality. However, at low bit rates, the performance of the generative model baseline significantly declines on this crucial downstream task of stereo matching, contrasting with the linear degradation observed in the VAE-based baseline.

Moreover, SDiD not only surpasses the state-of-the-art diffusion model RDEIC in LPIPS at ultra-low bit rates across both datasets but also shows consistent superiority in depth estimation tasks compared to the VAE baseline. Specifically, SDiD retains significantly more depth information in the decoded image pairs at a compression rate four times that of LDMIC (0.016 bpp vs 0.069 bpp), achieving an RMSE

of 3.99 compared to 4.11 for LDMIC.

Figure 5 presents the resultant visualizations. Compared to generative-based baselines, SDiD can faithfully restore image details while maintaining high realism and fidelity. Although the distortion optimization in CAMSIC yields higher PSNR, the distortions and noise introduced at ultra-low bit rates lead to a significant loss of texture details, making it difficult for downstream models to extract depth information from the compressed stereo images.

#### 4.4.2. CODING COMPLEXITY

We present the analysis of encoding and decoding complexity between SDiD and various baselines in the Table 1. Here, we standardize the input image size to $1024 \times 1024$, and the encoding/decoding time in the table refers to the average time for processing a single image. Benefiting from the fast sampling of DPA and the inter-view prior sharing strategy, the time cost brought by few-step diffusion is split equally between the two images, enabling SDiD to achieve a more satisfactory decoding speed than all diffusion baselines. Compared with RDEIC, which also adopts the few-step approach, SDiD still achieves an approximate 42.2% improvement in decoding speed.

### 4.5. Ablation studies

#### 4.5.1. MODULE ABLATIONS

We conducted ablation experiments on various modules we proposed, as detailed in the Table 3 (left). Using SDiD as the benchmark, we calculated the BD-rate of the ablation models in terms of LPIPS on the InStereo2K dataset. Discarding DPA significantly compromises performance, as it limits us to traditional DDPM sampling methods. Removing

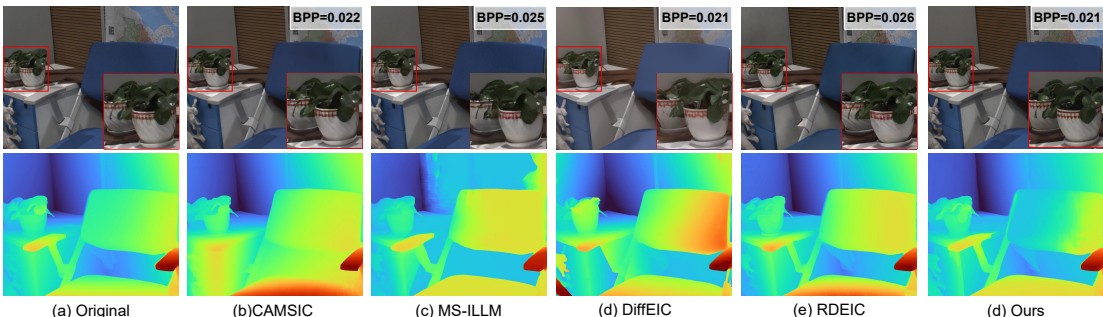

*Figure 5.* Quantitative comparison on the InStereo2K datasets.

*Table 1.* Encoding and decoding time evaluated on a pair of stereo images with the resolution as $1024 \times 1024$. The test results represent the time averaged per image.

| Models | LDMIC | CAMSIC | HIFIC | ILLM | Perco | DiffEIC | RDEIC | **SDiD(Ours)** |
|---|---|---|---|---|---|---|---|---|
| Encode Latency | 5.36s | 0.31s | 0.02s | 0.14s | 0.78s | 0.44s | 0.62s | **0.50s** |
| Decode Latency | 15.70s | 0.29s | 0.09s | 0.14s | 14.89s | 21.29s | 2.44s | **1.41s** |
| Decode NFE | 1 | 1 | 1 | 1 | 30 | 50 | 5 | **2** |

ANE restricts sampling to a Gaussian distribution, increasing the difficulty of cross-step sampling and resulting in a degradation of performance to some extent. Eliminating the DPT module prevents auxiliary views from leveraging the prior information of the foundational diffusion model, which has a devastating impact on reconstruction quality; this is qualitatively illustrated in Figure 5. It is noteworthy that both the DPT module and LBA are designed for auxiliary view decoding, so their removal does not affect the reconstruction results of the main-view.

#### 4.5.2. LOSS ABLATIONS

Table 3 (right) presents the ablation study on reconstruction losses. Both $\mathcal{L}_{adv}$ and $\mathcal{L}_{dino}$ can effectively enhance the realism of reconstructed images, and the $\mathcal{L}_{adv}$ can more significantly improve the depth estimation performance of decoded images. This indicates that maintaining the consistency between the reconstructed image and the original image in the feature domain of the self-supervised model is conducive to maximizing the retention of the semantic and depth information of the compressed image.

## 5. Conclusion

We propose SDiD, a distributed stereo image compression scheme based on sharing the prior of a foundation diffusion model. While addressing the difficulty of maintaining inter-view consistency in generative image compression, SDiD accelerates the sampling process through prior alignment. We have validated on multiple stereo image datasets that, under ultra-low bitrates, SDiD not only achieves superior realism and perceptual fidelity but also retains the depth

*Table 2.* BD-Rate (%) comparison with SOTA methods on Cityscapes and InStereo2K datasets. A larger BD-Rate indicates a greater performance gap compared to the reference model (SDiD).

| | InStereo2K | | | |
|---|---|---|---|---|
| Methods | Human Vision | | Machine Vision | |
| | LPIPS | DISTS | RMSE | 5pix |
| LDMIC | 783.01 | 593.82 | 271.95 | 220.93 |
| CAMSIC | 792.32 | 548.49 | 229.28 | 145.16 |
| HiFiC | 499.79 | 705.05 | 555.05 | 1012.27 |
| MS-ILLM | 97.77 | 237.96 | 276.94 | 406.44 |
| DiffEIC | 195.59 | 190.05 | 96.67 | 59.95 |
| RDEIC | 80.19 | 79.71 | 97.63 | 42.92 |
| **SDiD** | **0** | **0** | **0** | **0** |
| | Cityscapes | | | |
| Methods | Human Vision | | Machine Vision | |
| | LPIPS | DISTS | RMSE | 5pix |
| LDMIC | 255.01 | 693.22 | 33.13 | 47.99 |
| CAMSIC | 257.72 | 681.75 | 90.98 | 67.86 |
| HiFiC | 122.08 | 441.23 | 169.15 | 351.34 |
| MS-ILLM | 23.45 | 148.22 | 109.32 | 188.51 |
| DiffEIC | 117.86 | 141.92 | 154.51 | 70.83 |
| RDEIC | 25.07 | 57.63 | 27.99 | 31.34 |
| **SDiD** | **0** | **0** | **0** | **0** |

*Table 3.* Ablation studies comparing each module and reconstruction loss components on the InStereo2K dataset

| | BD-rate(%)(LPIPS) | | | BD-rate(%) | |
|---|---|---|---|---|---|
| Variant | Main_view | Aux_view | Variant | LPIPS | RMSE |
| w/o DPA | 113.52 | 152.15 | w/o $\mathcal{L}_{per}$ | 80.22 | 30.11 |
| w/o ANE | 21.53 | 25.20 | w/o $\mathcal{L}_{dino}$ | 15.31 | 35.26 |
| w/o DPT | 0.00 | 232.71 | w/o $\mathcal{L}_{adv}$ | 27.18 | 15.45 |
| w/o LBA | 0.00 | 25.50 | **Ours** | **0.00** | **0.00** |
| w/o Aux | 18.2 | 0.00 | | | |
| **Ours** | **0.00** | **0.00** | | | |

information of the original images to the greatest extent, facilitating depth estimation for downstream models. Overall, SDiD provides a practical stereo image compression framework for ultra-low bitrate scenarios.

## Impact statement

SDiD is a deep compression scheme built upon pre-trained diffusion models, applicable to distributed stereo image compression scenarios demanding high compression ratios. Similar to other generative models, it suffers from generation hallucinations and heavy computational overhead, so cautious deployment is required in scenarios that call for high-confidence outputs or have strict computing resource constraints.

## Acknowledgments

This work is supported in part by the National Natural Science Foundation of China under grant 624B2088, 62301189, 62571298, and Shenzhen Science and Technology Program under Grant KJZD20240903103702004, SYSPG20241211173609009.

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

## A. Proof

We assume that the compressed representation $\hat{c} = \mathcal{C}(z)$ is obtained from a compressor $\mathcal{C}(\cdot)$ based on the VAE architecture (Ballé et al., 2017). Then $\hat{c}$ is modeled as a Gaussian distribution $\hat{c} \sim \mathcal{N}(z, \sigma^2)$. Accordingly, the problem proposed in Section 3.3 can be transformed into finding a time step $t$ during the forward diffusion process, where the diffusion distribution $\mathcal{D}_{\mathrm{fd}}(t) \sim \mathcal{N}(a_t z, 1 - a_t^2)$ at this time step has the minimum distance to the distribution of the compressed representation:

$$\arg\min_t \quad \mathbf{dist}\left(\hat{c}, \mathcal{D}_{\mathrm{fd}}(t)\right). \tag{17}$$

Here, $\mathbf{dist}(\cdot, \cdot)$ denotes the distance metric. We adopt the Wasserstein Distance, and the minimization problem is converted into finding the minimum value of the following expression:

$$f(a_t) = (z - a_t z)^2 + \left(\sigma - \sqrt{1 - a_t^2}\right)^2. \tag{18}$$

We compute the minimum by setting the derivative to zero:

$$f'(a_t) = -2z^2(1 - a_t) + \frac{2a_t\left(\sigma - \sqrt{1 - a_t^2}\right)}{\sqrt{1 - a_t^2}} = 0. \tag{19}$$

Rearranging terms yields:

$$z^2 - a_t\left(z^2 - 1\right) = \frac{a_t \sigma}{\sqrt{1 - a_t^2}}. \tag{20}$$

According to the definition of signal-to-noise ratio, the SNR of $z$ is defined as $\mathrm{SNR}_{\hat{c}} = \frac{z^2}{\sigma^2}$, and the SNR of $\mathcal{D}_{\mathrm{fd}}(t)$ is: $\mathrm{SNR}_F(t) = \frac{a_t^2 z^2}{1 - a_t^2}$. The above equation can be rewritten as:

$$\frac{z^2(1 - a_t)}{a_t} + 1 = \frac{\sigma}{\sqrt{1 - a_t^2}} \rightarrow \frac{z^2(1 - a_t)}{a_t} + 1 = \frac{1}{a_t}\frac{\sigma}{z}\sqrt{\frac{a_t^2 z^2}{1 - a_t^2}} \rightarrow \frac{z^2(1 - a_t)}{a_t} + 1 = \frac{1}{a_t}\sqrt{\frac{\mathrm{SNR}_{F(t)}}{\mathrm{SNR}_{\hat{c}}}} \tag{21}$$

If we assume $z = \Bbbk$, the above equation reduces to:

$$\sqrt{\frac{\mathrm{SNR}_{F(t)}}{\mathrm{SNR}_{\hat{c}}}} = 1$$

This indicates that minimizing the Wasserstein Distance is fully consistent with the search objective of our ANE module.

Theoretically, the estimation of ANE is robust. We denote the theoretical minimum at this point as $a_t = a = \frac{1}{\sqrt{\sigma^2 + 1}}$. Since the timesteps in the forward process are discretized, the computed result of ANE will deviate from the true value. We denote the ANE-estimated value as $a_\star = a + \epsilon$, and compute $f(a_\star)$ using Taylor expansion:

$$f(a_\star) = f(a) + \frac{1}{2} \cdot \frac{2\left(1 + \sigma^2\right)^{3/2}}{\sigma^2} \cdot \epsilon^2 + o\left(\epsilon^2\right) \tag{22}$$

In low-bitrate scenarios, the variance $\sigma$ of the decoded image does not tend to zero, and the sensitivity of the overall error is $\Theta(\sigma) * o(\epsilon)$. Since the compressor $\mathcal{C}$ controls excessive variance via optimization objectives and clipping during training, the overall sensitivity of ANE is controllable.

## B. Setting Details

### B.1. Implementation Details

Our proposed method is implemented using PyTorch (Paszke et al., 2019). Experiments were conducted on two Intel(R) Xeon(R) Silver 4210 CPUs and single NVIDIA A6000 GPU. The Adam optimizer (Kingma & Ba, 2014) was employed with a learning rate of $0.5 \times 10^{-5}$.

To achieve compression at different bit rates in the training phase, we set the parameter $\lambda_1$ and $d_2$ to 3 , $d_1$ set to 0.2 and $d_3$ to 0.1, We then adjust $\lambda_2 \in \{6, 8, 12, 18\}$ to balance rate-distortion. This phase involved 80,000 training steps, allowing the model to sufficiently learn the underlying patterns and relationships necessary for effective compression across various bit rates.

The total training steps for the Instereo2K datasets were set at 50000, while the Cityscapes dataset was set at 80000. Across all datasets, a batch size of 2 was used. During the training process, these images were randomly cropped to a resolution of $512 \times 512$, a resolution that aligns with the input specifications of many advanced diffusion models, ensuring compatibility and effective feature learning. At the same time, Cityscape follows the conventional preprocessing approach: for every image, we crop 64, 256, and 128 pixels from the top, bottom, and sides, respectively, to remove the car hood (Wödlinger et al., 2022; Zhang et al.).During testing, we first pad the image to a multiple of 64, and then crop out the padded part after decoding.

### B.2. Baseline and Metrics Details

All the baseline approaches we utilized employed their respective official open-source codes[1][2]. To ensure result alignment, these baselines were retrained on Instereo2K and cityscapes, respectively. For the VAE-based methods LDMIC [3] and CAMSIC[4], since the bit rates they reported do not involve ultra-low bit rates, we used the code from their official open-source repositories, adjusted the rate-distortion hyperparameters, and retrained them on the corresponding datasets to obtain results with lower bit rates.

For DiffEIC[5], PerCo[6], RDEIC[7] we utilized the code and checkpoints provided by their official repositories.

For LPIPS, we utilized the `lpips` library, while DISTS was implemented using `DISTS_pytorch`. For the downstream depth estimation method `Foundation_stereo`, we utilized the checkpoints provided by their official repositories[8]. Additionally, we used the Bjntegaard delta metric (BD-metric) (Bjontegaard, 2001) to evaluate the gain in the corresponding metrics at the same bitrate level.

## C. Details of Algorithm Procedure

In this section, we supplement the explanation of the encoding and decoding processes of SDiD through pseudocode. The detailed encoding and decoding processes are shown in algorithm 1 and algorithm 2.

## D. Additional Experiment Results

We have visualized the results of bitrate allocation and the effect of shareable diffusion priors on different views, as shown in the Figure 6. With the denoising capability of the basic diffusion model, the main view image clearly restores real details. Although these priors are generated based on the main view, the DPT module we designed can effectively transfer them to the auxiliary views, significantly enhancing the texture reconstruction of the images. Meanwhile, the independent compressor also achieves satisfactory bitrate allocation results: the bitrate is concentrated in high-texture regions, and almost no bits are wasted in smooth regions.

---

[1]HiFiC:github.com/Justin-Tan/high-fidelity-generative-compression

[2]MS-ILLM: https://github.com/facebookresearch/NeuralCompression

[3]LDMIC : https://github.com/Xinjie-Q/LDMIC

[4]CAMSIC: https://github.com/Xinjie-Q/CAMSIC

[5]DiffEIC : https://github.com/huai-chang/DiffEIC

[6]Perco :https://github.com/Nikolai10/PerCo

[7]RDEIC: https://github.com/huai-chang/RDEIC

[8]`Foundation_stereo`: https://github.com/iKrishneel/foundation_stereo

---

**Algorithm 1** Encoding Process

---

**Input**: input image pair $(\boldsymbol{x}_m, \boldsymbol{x}_a)$, compressor encoder $\mathcal{M}^e(\cdot)$, compressor decoder $\mathcal{M}^d(\cdot)$, stable diffusion's encoder $\mathcal{E}(\cdot)$, Image captioning model $\mathbf{IC}(\cdot)$, quantization operation $Q$.

1: $\boldsymbol{z_{m;0}}, \boldsymbol{z_{a;0}} = \mathcal{E}(\boldsymbol{x}_m), \mathcal{E}(\boldsymbol{x}_a)$.
2: $\boldsymbol{y}_m = \mathcal{M}^e(\boldsymbol{z_{m;0}})$, $\boldsymbol{y}_a = \mathcal{M}^e(\boldsymbol{z_{a;0}})$.
3: $\hat{\boldsymbol{y}}_m = Q(\boldsymbol{y}_m)$, $\hat{\boldsymbol{y}}_a = Q(\boldsymbol{y}_a)$.
4: $\hat{\boldsymbol{c}}_m = \mathcal{M}^d(\boldsymbol{\hat{y}_m})$.
5: $\hat{t}(\hat{\boldsymbol{c}}_m) = \arg\min_t \|\mathrm{SNR}_F(t) - \mathrm{SNR}_{\hat{c}}\|_1$
6: Encode $\hat{\boldsymbol{y}}_m, \hat{\boldsymbol{y}}_a, \hat{t}(\hat{\boldsymbol{c}}_m)$ to binary file $\mathrm{bin}_\mathrm{x}$

**Output**: Binary file $\mathrm{bin}_\mathrm{x}$

---

**Algorithm 2** Decoding Process

---

**Input**: Binary file $\mathrm{bin}_\mathrm{x}$, compressor decoder $\mathcal{M}^d(\cdot)$, stable diffusion's decoder $\mathcal{D}(\cdot)$, Diffusion prior alignment module $\mathcal{A}_\phi(\cdot)$, Diffusion prior transfer module $\mathcal{PT}_\gamma(\cdot)$, sequence of timesteps $\tau_1 > \tau_2 > \cdots > \tau_{N-1}$. Noise schedule $\alpha(t), \beta(t)$.

1: Decode $\hat{\boldsymbol{y}}_m, \hat{\boldsymbol{y}}_a, \hat{t}(\hat{\boldsymbol{c}}_m)$ from binary file $\mathrm{bin}_\mathrm{x}$.
2: $\hat{\boldsymbol{c}}_m, = \mathcal{M}^d(\boldsymbol{\hat{y}_m})$, $\hat{\boldsymbol{c}}_a, = \mathcal{M}^d(\boldsymbol{\hat{y}_a})$.
3: $\hat{\boldsymbol{z}}_{m;0} = \frac{\left(\hat{\boldsymbol{c}}_m - \sqrt{1-\bar{\alpha}(\hat{t}(\hat{\boldsymbol{c}}_m))}\epsilon_\theta^t\right)}{\sqrt{\bar{\alpha}(\hat{t}(\hat{\boldsymbol{c}}_m))}}$
4: $\tilde{\boldsymbol{z}}_{m;0} \leftarrow \mathcal{A}_\phi\left(\hat{\boldsymbol{z}}_{m;0}, \hat{\boldsymbol{c}}_m, \hat{\boldsymbol{c}}_a, \hat{t}(\hat{\boldsymbol{c}}_m)\right)$
5: $\tilde{\boldsymbol{z}}_{a;0} \leftarrow \mathcal{PT}_\gamma\left(\tilde{\boldsymbol{z}}_{m;0}, \hat{\boldsymbol{c}}_m, \hat{\boldsymbol{c}}_a, \hat{t}(\hat{\boldsymbol{c}}_m)\right)$
6: **for** $n = 2$ to $N - 1$ **do**
7:     $\boldsymbol{z}_{\tau_n} \sim \mathcal{N}\left(\alpha\left(\tau_n\right)\tilde{\boldsymbol{z}}_0; \beta^2\left(\tau_n\right)\boldsymbol{I}\right)$
8:     $\tilde{\boldsymbol{z}}_{m;0} \leftarrow \mathcal{A}_\phi\left(\hat{\boldsymbol{c}}_m, \hat{\boldsymbol{c}}_a, \tau_n\right)$
9:     $\tilde{\boldsymbol{z}}_{a;0} \leftarrow \mathcal{PT}_\gamma\left(\tilde{\boldsymbol{z}}_{m;0}, \hat{\boldsymbol{c}}_m, \hat{\boldsymbol{c}}_a, \tau_n\right)$
10: **end for**
11: $\hat{\boldsymbol{x}}_m = \mathcal{D}(\tilde{\boldsymbol{z}}_{\boldsymbol{m;0}})$, $\hat{\boldsymbol{x}}_a = \mathcal{D}(\tilde{\boldsymbol{z}}_{\boldsymbol{a;0}})$.

**Output**: Decoded image $\hat{\boldsymbol{x}}_m, \hat{\boldsymbol{x}}_a$

---

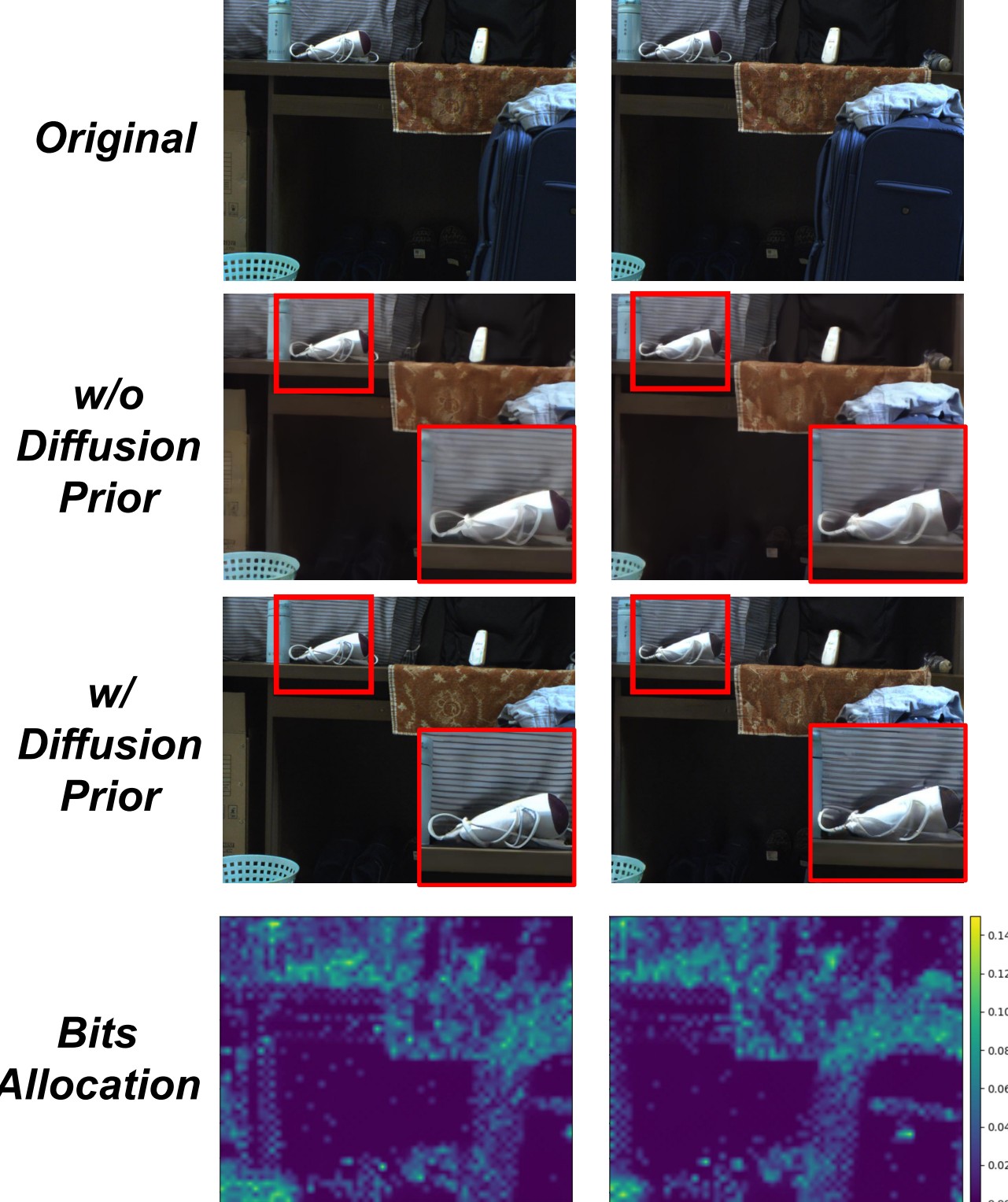

*Figure 6.* Ablation experiments on diffusion priors and visualization of bitrate allocation

