# OpenReview forum: "SDiD：Shared diffusion prior for efficient distributed stereo image compression"
_ICML.cc/2026/Conference — ICML 2026 regular_

### Official Review · Reviewer_4ymr · 2026-03-01

**Soundness:** 2
**Presentation:** 2
**Significance:** 3
**Originality:** 2
**Overall Recommendation:** 4
**Confidence:** 2

**Summary:**

This paper proposes SDiD, a distributed stereo image compression framework.
During encoding, it leverages a shared latent compressor to obtain compressed latents.
The decoding includes DPA for prior alignment, DPT for transferring the main-view latent to the auxiliary view,  and ANE for adaptive timestep estimation.
Experiments show strong gains on perceptual quality and downstream depth estimation.

**Compliance With Llm Reviewing Policy:**

Affirmed.

**Final Justification:**

The added experiments in rebuttal addressed my concerns.

**Key Questions For Authors:**

Why not comparing your method aginest ECSIC [Matthias Wödlinger, et al. WACV 2024] and BiSIC [Zhening Liu, et al. ECCV 2024]?

Other concerns are specified in Weaknesses.

**Limitations:**

The reviewer did not see the limitations or the societal impact in this submission.

**Strengths And Weaknesses:**

Strengths
- The paper studies an important problem. For stereo image compression, cross-view consistency is crucial. And the overall method is interesting.
- The experimental results are strong, on both perceptual quality and depth estimation.

Weaknesses
- Comparisons omit several representative and open-sourced baselines: ECSIC [Matthias Wödlinger, et al. WACV 2024] and BiSIC [Zhening Liu, et al. ECCV 2024].
- DiffEIC [Zhiyuan Li, et al. TCSVT 2025], RDEIC [Zhiyuan Li, et al. TCSVT 2025], PerCo [Marlène Careil, et al. ICLR 2024] are single-image compression methods based on diffusion models, the better performance over them can be expected.
- The writing has some artifacts. For example in the DPA part of Figure 2, the arrow of $\hat c_m$ is wrong and the arrow of $\hat c_a$ is missing.

---

> ### Author Rebuttal · Authors · 2026-03-31
>
> We greatly appreciate your comments. We are highly encouraged by your recognition of the superior performance of our method and the significance of the problem we address. We will make every effort to address your questions.
>
> **Comparisons omit several representative and open-sourced baselines**
>
> Thank you for your reminder! Following your suggestion, we have tested these methods using their publicly available pre-trained weights and compared them with SDiD.
> These baselines perform worse than the multi-view baseline CAMSIC already compared in our paper, and thus are significantly outperformed by our proposed SDiD, as shown in the table below:
>
> | Instero2k   | Bpp   | PSNR    | LPIPS | DISTS |
> | ----------- | ----- | ------- | ----- | ----- |
> | SDiD (Ours) | 0.016 | 27.0141 | 0.151 | 0.103 |
> |             | 0.024 | 28.321  | 0.127 | 0.087 |
> |             | 0.035 | 29.211  | 0.112 | 0.073 |
> |             | 0.052 | 30.251  | 0.094 | 0.062 |
> |             |       |         |       |       |
> | ECSIC       | 0.032 | 32.372  | 0.295 | 0.202 |
> |             | 0.044 | 33.391  | 0.264 | 0.178 |
> |             | 0.069 | 34.620  | 0.226 | 0.148 |
> |             |       |         |       |       |
> | BiSIC       | 0.214 | 37.470  | 0.105 | 0.071 |
>
> It can be observed that compared with ECSIC, SDiD still achieves superior perceptual reconstruction quality even with only one-fifth of the bitrate (0.016 bpp vs. 0.069 bpp).
> Compared with BiSIC, SDiD only requires a quarter of the bitrate (0.052 bpp vs. 0.214 bpp) while delivering better perceptual reconstruction quality.
> We also present performance comparisons of these baselines on the Cityscapes dataset in https://anonymous.4open.science/r/anonymous-git-EEA2/README.md (Table 7)
>
> These additional baseline results do not affect the conclusion that SDiD achieves state-of-the-art performance at low bitrates.
>
> **The writing has some artifacts:**
> Thank you for your careful reading! We apologize for the reading difficulties caused by our oversight. We will correct these errors promptly in the revised version, carefully check all details, and further improve the readability of the manuscript.
>
> Once again, we appreciate your valuable suggestions. We are willing to clarify any further questions to help refine our work. Additional experimental results are presented in https://anonymous.4open.science/r/anonymous-git-EEA2/README.md, and you may also refer to our discussions with other reviewers for more details.
>
>
> We would be grateful if you could consider raising your rating, provided that our responses have adequately addressed your concerns.

---

> > ### Author Rebuttal · Reviewer_4ymr · 2026-04-03
> >
> > Thanks for the rebuttal. The added experiments are clear. I will give weak accept.

---

> > > ### Author Response · Authors · 2026-04-03
> > >
> > > Thank you for your review！We appreciate your positive feedback on our work.

---

### Official Review · Reviewer_X1TH · 2026-03-02

**Soundness:** 2
**Presentation:** 2
**Significance:** 3
**Originality:** 3
**Overall Recommendation:** 4
**Confidence:** 3

**Summary:**

This paper proposes a distributed stereo image compression method SDiD based on shared diffusion prior, which aims to solve the trade-off between view consistency and generation quality in stereo image compression at very low bit rates. The method mainly includes three modules: DPA, ANE and DPT. The implementation is carried out on Cityscapes and InStereo2K datasets. The results show that SDiD outperforms the existing methods in terms of perceptual quality, and the decoding speed is significantly improved.

**Compliance With Llm Reviewing Policy:**

Affirmed.

**Final Justification:**

Thanks for the rebuttal. The added experiments are clear. I will give weak accept.

**Key Questions For Authors:**

1.How can the training stability of DPA modules be guaranteed? The article mentions that Aϕ is initialized to zero, is there a problem of instability or mode collapse at the beginning of training? Are other training techniques used?

2.How much does the estimation accuracy of ANE affect the decoding quality? Figure 3 demonstrates the estimation effect of ANE, but does not quantify the impact of estimation errors on the final reconstruction quality. It is suggested to supplement the sensitivity analysis, presented  tˆ effect of bias on LPIPS/RMSE.

3.Does the LBA in the DPT module introduce additional computational overhead? Does LBA have an advantage in computational efficiency and memory usage over standard Cross-Attention? It is suggested to supplement the complexity comparison.

**Limitations:**

YES

**Strengths And Weaknesses:**

Strengths：This paper proposes a distributed stereo image compression method SDiD based on shared diffusion prior, which aims to solve the trade-off between view consistency and generation quality in stereo image compression at very low bit rates. The method mainly includes three modules: DPA, ANE and DPT. The implementation is carried out on Cityscapes and InStereo2K datasets. The results show that SDiD outperforms the existing methods in terms of perceptual quality, and the decoding speed is significantly improved.


Weaknesses：There are problems with the typesetting of the article, making it difficult to read. There is a lack of theoretical guarantee analysis for the convergence of the DPA module, the estimation error of the ANE, etc. The overall model contains multiple modules, and there is no comparison with existing methods in memory, running time and running efficiency.

---

> ### Author Rebuttal · Authors · 2026-03-31
>
> We sincerely appreciate your comments. We are greatly encouraged by your recognition of the superior performance of our proposed method. We will do our utmost to address your concerns:
>
> **Comparison with existing methods in memory, running time, and running efficiency:** Thank you for your constructive suggestions!
> We present the number of parameters and computational complexity of SDiD in Table 3 in https://anonymous.4open.science/r/anonymous-git-EEA2/README.md, along with comparisons to baseline methods.
> You may also refer to the experiments we conducted and our discussions with Reviewer Gwkd and Reviewer Debk.
>
> **DPA training instability:** Thank you for your comment. We considered training stability when designing the DPA module. In fact, DPA adopts a residual structure to ensure stable training, as illustrated by the yellow part in Figure 2. The purpose of the DPA is to align the diffusion prior estimate $\hat{z}^t _0$ with the main-view compressed representation $\hat{c} _m$, to obtain a more accurate estimation of $z _0$ (cf. Equation 7 in the original paper).
>
> Since both $\hat{z}^t _0$ and $\hat{c} _m$ are essentially estimates of the true representation $z_0$, the loss can be relatively well controlled in the early training stage even with random initialization. From another perspective, the weights of the pre-trained diffusion network are frozen, so DPA only serves to calibrate the prior during training and does not participate in the generation of specific patterns. This results in relatively low training difficulty and thus ensures a highly stable training process.
>
>
> **Estimation errors on the final reconstruction quality:** The purpose of the ANE module is to estimate the noise level of the distorted compressed representation $\hat{c}$, to infer the most similar intermediate sampling state $z _{\hat{t}}$.
> It should be noted that this prediction process has no ground-truth for reference or supervision. Therefore, we designed the following experiments to verify the prediction sensitivity of ANE:
>
> We conducted 7 groups of experiments on InStereo2K using different timesteps $t$, including $\hat{t}$ (the estimate from the ANE module) and $\hat{t}\pm i$ with $i=10,25,50$.
> We report the average reconstruction performance obtained using these different timesteps as follows:
>
> | $\hat{t} + k$ | k =-50 | k=-25 | k=-10 | k=0 ($\hat{t}$) | k=10  | k=25  | k=50  |
> | ------------- | ------ | ----- | ----- | --------------- | ----- | ----- | ----- |
> | bd_lpips      | 0.019  | 0.013 | 0.012 | 0               | 0.008 | 0.015 | 0.021 |
> | bd_dists      | 0.013  | 0.011 | 0.009 | 0               | 0.004 | 0.008 | 0.012 |
>
> The metric used for evaluation is **bd_metric**, where the estimate $\hat{t}$ from the ANE module serves as the baseline. A **larger value** indicates **worse performance**.
>
> It can be observed that the estimation $\hat{t}$ from ANE achieves the best accuracy compared with its neighboring timesteps. In addition, slight perturbations around $\hat{t}$ do not lead to significant degradation in reconstruction quality, which further verifies the robustness of the ANE module.
>
> **Complexity comparison of LBA:** We appreciate your constructive comment! We have compared the LBA module with standard cross-attention in terms of number of parameters and inference latency, as shown in the table below:
> | Module    | LBA       | Cross-attn |
> | --------- | --------- | ---------- |
> | Parameter | 115.07 KB | 115.06 KB  |
> | Latency   | 4.72ms    | 2.88ms     |
>
> Here we set the input resolution of the model to 512×512.
> Since LBA shares the same network structure as standard cross-attention, it introduces no extra parameters.
> However, LBA requires one additional convolution operation to embed $\hat{c} _m$, resulting in a slight increase in inference latency.
> Overall, the extra complexity introduced by LBA is controllable and negligible.
>
> Once again, we sincerely appreciate your valuable suggestions. We are happy to clarify any further questions to help improve our work.
> We would be grateful if you could consider raising your rating, provided that our responses have adequately addressed your concerns.

---

> > ### Author Rebuttal · Reviewer_X1TH · 2026-04-02
> >
> > Thank you for the detailed rebuttal. The additional clarifications are helpful. However, my main concerns are only partially addressed. In particular, it is not enough to explain the stability of Q2 only by residual. Q3 is more like some theoretical derivation, and it is not logical to prove it only by experimental results.
> >
> > Therefore, my overall assessment remains unchanged at this stage.

---

> > > ### Author Response · Authors · 2026-04-02
> > >
> > > Thank you for your reply. We hope to address your concerns as fully as possible:
> > >
> > > **Q 2**:  We analyze that the training stability of $A _{\phi}$ arises from three aspects:
> > >
> > > 1. **Input stability**. Unlike the generation paradigms of diffusion models and GANs, the inputs $\hat{z}^t _0$ and $\hat{c}$ received by $A _{\phi}$ are themselves estimates of $z$, differing only in their estimation patterns. The goal of $A _\phi$ is to align these two estimates and unify their patterns. This is fundamentally different from generative paradigms in terms of training difficulty and paradigm: the latter simulates a generation process from 0 to 1, while the former focuses on refinement and alignment.
> > >
> > > 2. **Structural stability**: As mentioned previously, the residual structure contributes to training stability to a certain extent.
> > >
> > > 3. **Training scheme**: For training, we adopt warmup and cosine annealing strategies. Such learning rate scheduling methods are commonly used in the training and fine-tuning of diffusion models.
> > >
> > > In summary, we do not propose a novel training scheme specifically for training stability; the model maintains satisfactory stability using existing mature training strategies.
> > >
> > > **Q 3**:
> > > Sorry for misunderstanding your meaning. We will elaborate on the error of this prediction theoretically as much as possible:
> > >
> > > Let the compressor be $\mathcal{C}(\cdot)$. Existing compressors all follow Balle's VAE [1] architecture in their framework, so $\hat{c}=\mathcal{C}(z)$ follows a Gaussian distribution $\hat{c}\sim \mathcal{N}(z,\sigma^2)$. (Here, $z$ corresponds to $z_0$ in Equation 3 of the original text.)
> > >
> > > In the forward process of Gaussian diffusion, for a fixed timestep $t$, the diffusion distribution at this state is $\mathcal{D}_\text{fd}(t)\sim \mathcal{N}(a_tz,1-a^2_t)$ (where $a_t=\sqrt{\bar{\alpha}_t}$, which is essentially consistent with Equation 3 in the original text and is used here for simplified description). The ANE is designed to:
> > >
> > > $$
> > > \underset{t}{\arg \min } \quad \textbf{dist}(\hat{c},\mathcal{D_fd}(t))
> > > $$
> > >
> > > Here $\textbf{dist}()$ is a distance metric. We use the Wasserstein Distance here, and this minimization problem is transformed into finding the minimum value of:
> > > $$
> > > f(a_t)=(z-a_t z)^2+\left(\sigma-\sqrt{1-a_t^2}\right)^2
> > > $$
> > >
> > > We compute the minimum by setting the derivative to zero:
> > > $$
> > > \begin{aligned}
> > > f^{\prime}(a_t)=-2 z^2(1-a_t)+\frac{2 a_t\left(\sigma-\sqrt{1-a_t^2}\right)}{\sqrt{1-a_t^2}}=0
> > > \end{aligned}
> > > $$
> > > Rearranging terms yields:
> > > $$
> > > z^2-a_t\left(z^2-1\right)=\frac{a_t \sigma}{\sqrt{1-a_t^2}}
> > > $$
> > >
> > > According to the definition of signal-to-noise ratio, the SNR of $z$ is defined as $SNR_{\hat{c}}=\frac{z^2}{\sigma^2}$, and the SNR of $\mathcal{D_fd}(t)$ is: $\operatorname{SNR}_F(t)=\frac{a_t^2z^2}{1-a_t^2}$. The above equation can be rewritten as:
> > >
> > > $$
> > > \begin{aligned}
> > >  \frac{z^2(1-a_t)}{a_t}+1=\frac{\sigma}{\sqrt{1-a_t^2}} \rightarrow \\\
> > > \frac{z^2(1-a_t)}{a_t}+1=\frac{1}{a_t}\frac{\sigma}{z}\sqrt{\frac{a_t^2z^2}{1-a_t^2}}\rightarrow\frac{z^2(1-a_t)}{a_t}+1=\frac{1}{a_t} \sqrt{\frac{S N R_{F(t)}}{S N R_{\hat{c}}}}
> > > \end{aligned}$$
> > > If we assume $z=\mathbb{1}$, the above equation reduces to:
> > > $$
> > > \sqrt{\frac{S N R_{F(t)}}{S N R_{\hat{c}}}}=1
> > > $$
> > > This indicates that minimizing the Wasserstein Distance is fully consistent with the search objective of our ANE module.
> > > We denote the theoretical minimum at this point as $a_t=a=\frac{1}{\sqrt{\sigma^2+1}}$.
> > > Since the timesteps in the forward process are discretized, the computed result of ANE will deviate from the true value.
> > > We denote the ANE-estimated value as $a_{\star}=a+\epsilon$, and compute $f(a_{\star})$ using Taylor expansion:
> > > $$
> > > f\left(a_{\star}\right) =f (a)+\frac{1}{2} \cdot \frac{2\left(1+\sigma^2\right)^{3 / 2}}{\sigma^2} \cdot \epsilon^2+o\left(\epsilon^2\right)
> > > $$
> > > In low-bitrate scenarios, the variance $\sigma$ of the decoded image does not tend to zero, and the sensitivity of the overall error is $\Theta(\sigma)*o(\epsilon)$.
> > > Since the compressor $\mathcal{C}$ controls excessive variance via optimization objectives and clipping during training, the overall sensitivity of ANE is controllable.
> > >
> > > Furthermore, if $z$ is not equal to 1, we use the above signal-to-noise ratio equation for estimation. The error relative to the theoretically minimal Wasserstein Distance is as follows:
> > > $$
> > >  \Delta \approx \frac{\sigma^2(\sqrt{\sigma^2+1}-1)^2}{(\sigma^2+1)^{5 / 2}}\left(z^2-1\right)^2
> > > $$
> > > In practice, $z$ is normalized to the range $[-1,1]$, so the above error is controllable. Moreover, this error will be further reduced as the variance of the decoded features from the compressor increases with decreasing bitrate.
> > >
> > > In the subsequent revision, we will unify the variable names throughout the above derivations and present them in the appendix.
> > > We sincerely appreciate your suggestions. Regardless of whether you decide to raise your rating, your review has helped us improve our work.
> > >
> > > [1] End-to-end Optimized Image Compression

---

### Official Review · Reviewer_Gwkd · 2026-03-05

**Soundness:** 3
**Presentation:** 2
**Significance:** 3
**Originality:** 2
**Overall Recommendation:** 4
**Confidence:** 4

**Summary:**

This paper presents SDiD, a stereo image compression framework utilizing a shared pre-trained diffusion prior to balance perceptual quality and cross-view consistency at ultra-low bitrates. Key contributions include: Diffusion Prior Alignment (DPA) for efficient two-step sampling via consistency objectives; an Adaptive Noise Estimator (ANE) for SNR-matched timestep selection; and a Diffusion Prior Transfer (DPT) module with Look-Back Attention for cross-view prior adaptation. Evaluations on InStereo2K and Cityscapes show that SDiD outperforms VAE, GAN, and single-image diffusion baselines in both perceptual quality and depth estimation accuracy, with faster decoding speeds than comparable few-step diffusion methods.

**Compliance With Llm Reviewing Policy:**

Affirmed.

**Key Questions For Authors:**

Could the authors provide a more complete mathematical derivation for the DPA module? Specifically, how do the input arguments of A_ϕ align across Equation 5 and Algorithm 2? It would also be helpful to explicitly write out the mapping from the noise prediction ϵ-to-z_0 to better understand its connection to standard diffusion formulations.

The paper heavily emphasizes its superior performance in "extremely low bitrate scenarios," noting that SDiD at 0.016 bpp retains better depth information than LDMIC at 0.069 bpp.
First, could the authors provide a deeper mechanistic explanation or intermediate feature visualizations showing why the geometric consistency (and thus downstream depth estimation accuracy) is so remarkably preserved by the DPT module, despite the inherent hallucination tendencies of diffusion models at such extreme compression levels?
Second, since ultra-low bitrate is positioned as a primary operating regime and advantage, comparing primarily against VAE-based methods (which inherently struggle at such bitrates) may not fully reflect the state-of-the-art. Could the authors discuss or compare SDiD against specialized ultra-low bitrate generative compression baselines to more rigorously validate this specific claim?

Table 1 reports the encoding and decoding latency based on a 1024*1024 image resolution. Since inference time can vary significantly across different hardware and implementation optimizations, could you supplement this with hardware-agnostic complexity metrics, such as FLOPs and parameter counts? Furthermore, please clarify if the diffusion-based baselines reported in Table 1 use the same number of sampling steps as the proposed SDiD.

Tables 2 and 3 report performance primarily using relative BD-Rate percentages. To help readers better contextualize these relative gains, could the authors provide a supplementary table showing the absolute evaluation values (e.g., raw LPIPS, DISTS, PSNR, MS-SSIM, and RMSE) at comparable bitrates for both the proposed method and the evaluated baselines?

**Limitations:**

The authors have not adequately discussed the limitations and potential negative societal impacts of their work, and a dedicated section for this should be included in the final manuscript. Since the paper motivates the need for efficient stereo image compression by referencing critical domains like autonomous driving, it is crucial to discuss the inherent risks of generative compression at ultra-low bitrates. Specifically, there is a distinct risk that diffusion models might hallucinate plausible but entirely fictitious visual details or geometric structures, which could mislead downstream algorithms in real-world deployments.

**Strengths And Weaknesses:**

The DPA module creatively adapts a frozen diffusion model via consistency regularization, enabling direct Z0 prediction and drastically reducing sampling steps. The ANE module introduces a lightweight, SNR-matching mechanism to dynamically determine starting timesteps, optimizing the speed-accuracy trade-off without additional training overhead.

The mathematical details of the DPA module require further clarification. The input variables for the mapping A_ϕ appear to vary across the text; for instance, Equation 5 includes ϵ_θ^t, while Algorithm 2 does not. Additionally, the specific mathematical relation to standard ϵ-to-z_0 conversions in diffusion models is not explicitly provided. Equation 6 needs more explanation regarding the target network's EMA mechanics and the boundary condition stabilization in Equation 7.

---

> ### Author Rebuttal · Authors · 2026-03-31
>
> We would like to thank the reviewer for their insightful and detailed suggestions. We also appreciate your recognition of the technical merits and innovation of our work.
>
> **Provide a more complete mathematical derivation for the DPA module**：
> In practice, for the main-view compressed representation $\hat{c}_m$, we first employ a controllable pre-trained denoising network (the red region in Fig. 2), which serves as the condition (along with the time embedding $t$) to generate the diffusion prior $ \epsilon^t _{\theta}$.
>
> Since this prior fits the Gaussian noise at time $t$, we first obtain an estimate of the true distribution via $\epsilon$-prediction:
>
> $$ \left(z_ t-\sqrt{1-\bar{\alpha} _t} \epsilon^t _ {\theta} \right) / \sqrt{\bar{\alpha} _t } = \hat{z} _{0}^t \sim z_0$$
>
> This estimate is coarse and unaligned. We then further refine it through the prior alignment module (the yellow region in Fig. 2):
>
> $$
> \mathcal{A} _\phi \left(\hat{z} _0^t, \hat{c} _m, \hat{c} _a, t\right)=\tilde{z} _{0 ;m} \sim z _{0 ; m}
> $$
>
> As $\hat{z}^t _0$ is mathematically equivalent to $\epsilon^t _{\theta}$, we use $\epsilon^t _{\theta}$ in Eq. 5 to directly emphasize the role of the diffusion prior in the module. Moreover, since $\epsilon^t _{\theta}$ in fact only takes $\hat{c} _m$ as an input variable, we further omit it in Algorithm 2.
>
> However, we recognize that such omissions in Algorithm 2 have compromised readability and made the algorithmic procedure harder to follow. We have decided to revise the presentation of Algorithm 2 in the revised version so that the transformation of intermediate variables is clearly illustrated. In the revised manuscript, we will also add detailed descriptions of the DPA module and $\epsilon$-pred in the main text.
>
> **Provide a deeper mechanistic explanation:** Pre-trained generative models suffer from inherent hallucinations, making it hard to ensure cross-view consistency by simply changing network structures or training objectives. The DPT module directly transfers and strengthens generative priors across views; with proper inter-view relationships, hallucinations remain geometrically consistent, reducing depth information loss during compression.
> By contrast, VAE-based multi-view approaches and other generative baselines either lose rich textures and geometry, impairing downstream depth estimation, or produce severe depth errors due to inconsistent cross-view hallucinations.
>
> We plan to add a certain number of visualization results to corroborate this phenomenon in future versions.
>
> **Discuss or compare SDiD against specialized ultra-low bitrate generative compression baselines:**  In practice, we have compared many generative baselines in the main text, such as HiFiC, DiffEIC, and RDEIC. All these methods are built upon GAN or diffusion-based generative models and achieve excellent performance at extremely low bitrates. SDiD also exhibits significant advantages over these approaches. For further validation, we compare our method with CDC on the Instereo2K dataset.
>
> | | Bpp| LPIPS| DISTS | 5-pixel |
> | --| --| --| -- | -- |
> | CDC  | 0.13| 0.195| 0.13| 0.09|
> | SDiD | 0.02| 0.127| 0.08| 0.06|
>
> SDiD achieves comprehensive performance advantages using only one-sixth the bitrate of CDC.
>
> **FLOPs and parameter counts:**  We compare the computational complexity and number of parameters between SDiD and the baseline methods separately for encoding and decoding in Table 3, available via Link https://anonymous.4open.science/r/anonymous-git-EEA2/README.md.
>
> SDiD exhibits clear advantages in encoding complexity compared with DiffEIC, which also adopts a diffusion architecture, and remains competitive against the VAE-based method CAMSIC.
> For the decoding stage, SDiD has a relatively large number of parameters and high complexity. However, thanks to prior sharing and a reduced sampling mechanism, it still requires significantly less computation than other diffusion-based frameworks.
> Notably, in distributed scenarios, the decoder is often deployed on the cloud, so the elevated computational overhead is relatively tolerable.
>
>
> **Clarification on the sampling steps of diffusion‑based baselines reported in Table 1.**
> The Number of Function Evaluations (NFE) for these diffusion baselines is shown in the table below:
>
> | | Perco | DiffEIC | RDEIC | SDiD |
> | ---- | ----- | ------- | ----- | ---- |
> | NFE  | 30| 50 | 5 | 2|
>
> We will add the NFE values for different methods to Table 1 in the revised version.
>
> **Provide absolute evaluation values**
> Due to the word limit in this response, we present all absolute values in Table 4-7 in https://anonymous.4open.science/r/anonymous-git-EEA2/README.md. We will also include them in the appendix in the revised version.
>
> **Discussed the limitations**
> We will add a new section in the revised version to discuss the limitations of our work and its potential societal impacts in detail.

---

> > ### Author Rebuttal · Reviewer_Gwkd · 2026-04-03
> >
> > Thanks for the rebuttal.

---

> > > ### Author Response · Authors · 2026-04-03
> > >
> > > Thank you for your appreciation of our work! We have also added theoretical proofs regarding ANE during our discussions with other reviewers, which you may refer to at your convenience if you are interested.

---

### Official Review · Reviewer_Debk · 2026-03-05

**Soundness:** 3
**Presentation:** 3
**Significance:** 3
**Originality:** 3
**Overall Recommendation:** 4
**Confidence:** 4

**Summary:**

This paper proposes Shared Diffusion Prior for Efficient Distributed Stereo Image Compression (SDiD) to address the precision loss in downstream tasks caused by cross-view inconsistency and the exorbitant denoising costs. The main technical innovations include: 1) A Diffusion Prior Alignment (DPA) module equipped with Adaptive Noise Estimation (ANE) to gauge the noise level of compressed representations and align the pre-trained prior with the latent space, enabling the rapid generation of the main-view prior through ultra-fast two-step sampling. 2) A Diffusion Prior Transfer (DPT) module incorporating Look-Back Attention (LBA) that explicitly captures correlations between inter-view representations, ensuring cross-view consistency. Comprehensive experimental results demonstrate the effectiveness of SDiD.

**Compliance With Llm Reviewing Policy:**

Affirmed.

**Final Justification:**

The rebuttal has addressed my concerns. I give my score of 4.

**Key Questions For Authors:**

1. Could the authors provide more concrete mathematical formulations or architectural details for the Diffusion Prior Transfer (DPT) module and Look-Back Attention (LBA)? This would help improve readers’ understanding and facilitate reproducibility.
2. Since the encoding latency of SDiD (0.50 s) is noticeably higher than that of some joint coding baseline models, the authors are encouraged to report the computational cost (e.g., FLOPs) to support the claim that the encoder is relatively lightweight for deployment on edge devices.

**Limitations:**

No. The authors should discuss potential failure scenarios of the proposed algorithm and its possible negative societal impacts.

**Strengths And Weaknesses:**

Strengths:
1.	Clear research motivation: The paper clearly identifies key challenges in stereo image compression, including precision loss and high denoising cost, which are particularly relevant for asymmetric distributed coding scenarios.
2.	Certain degree of innovation: The proposed SDiD framework shares the main-view diffusion prior with the auxiliary view via a DPT module and introduces a two-step sampling strategy, providing a novel application of diffusion techniques in stereo compression.
3.	Technical soundness: The proposed framework for distributed stereo image compression is well designed and technically feasible.
4.	Comprehensive evaluation: Through comparisons with various state-of-the-art methods and ablation studies, the authors provide a comprehensive evaluation.

Weaknesses:
1.	The introduction explicitly targets scenarios like autonomous driving, which typically require real-time processing. Although the diffusion process is accelerated, the latency still appears too high for low-latency applications. The authors are encouraged to discuss possible future directions to further reduce latency.
2.	The experiments are conducted only on high-resolution datasets. It would be beneficial to include more diverse datasets to evaluate the robustness of SDiD across different resolutions.
3.	The paper has several presentation issues affecting readability:
1)	The background section in the abstract is overly long and fails to clearly highlight the core innovative modules of the proposed algorithm.
2)	In the introduction, it is recommended to organize the overall algorithm workflow more clearly around the innovations and principles of each module.

---

> ### Author Rebuttal · Authors · 2026-03-31
>
> We would like to express our sincere gratitude to the reviewer for the insightful and constructive comments! We are greatly encouraged by your recognition of the technical merits and innovation of our work, and we will answer your questions to the best of our ability:
>
> **The authors are encouraged to discuss possible future directions:**
> As revisions to the manuscript are not feasible at this stage, we will elaborate on future research directions and existing limitations in the Conclusion section in subsequent versions. Overall, we believe that future work should focus on constructing lightweight and efficient encoder, and aligning the inputs of these encoders with the analytical encoders of pre-trained latent diffusion models in the feature domain, so as to enable real edge device deployment and high-fidelity decoding.
>
> **Evaluate the robustness of SDiD across different resolutions:**
> Since most existing public stereo image datasets are in high resolution, we downsampled the InStereo2k and Cityscapes datasets and retested SDiD as well as several baseline methods. We downsampled the images from both datasets to 512×512 pixels and the lower resolution of 256×256 pixels, respectively. Partial experimental results are shown below:
>
> | MODEL （256） | Bpp   | Lpips | Dists |
> | ------------- | ----- | ----- | ----- |
> | LDMIC         | 0.089 | 0.306 | 0.235 |
> | SDiD (Ours)   | 0.032 | 0.188 | 0.127 |
>
> Due to space limitations, please refer to Table 1-2 in the anonymous repository https://anonymous.4open.science/r/anonymous-git-EEA2/README.md for the complete experimental results.
>
> **The paper has several presentation issues affecting readability:** We appreciate your valuable comments. We will conduct a comprehensive revision of the manuscript in accordance with your suggestions at the stage when modifications to the original text are permitted. In the Introduction section, we will add additional paragraphs to provide a detailed explanation of the overall algorithm pipeline to improve readability.
>
> **Mathematical formulations or architectural details for the DPT module and LBA:** We will add more detailed descriptions in the method section. The main function of the DPT module is to align the diffusion prior used for the main view to the auxiliary view. First, we employ the DPA module $\mathcal{A} _\phi(\cdot)$ to enhance the main-view representation using the generative prior:
>
> $$
> \mathcal{A} _\phi\left(\boldsymbol{\epsilon} _{\boldsymbol{\theta}}^{\boldsymbol{t}}, \hat{c} _m, \hat{c} _a, t\right)=\tilde{z} _{\mathbf{0} ; \boldsymbol{m}}\sim z _{\mathbf{0} ; \boldsymbol{m}}
> $$
>
> Subsequently, the DPT module $\mathcal{P} \mathcal{T} _\gamma(\cdot)$ efficiently aligns the diffusion prior from the main view to the auxiliary view:
>
> $$
> \mathcal{P} \mathcal{T} _\gamma\left(\tilde{z} _{ \textbf{0};\boldsymbol{m} }, \hat{c} _m, \hat{c} _a, \tau _n\right)=\tilde{z} _{ \textbf{0};\boldsymbol{a} }\sim z _{\mathbf{0} ; \boldsymbol{a}}
> $$
>
> In DPT, we adopt LBA, an effective alignment module based on cross-attention:
>
> $$
> \operatorname{LBA}\left(\hat{\boldsymbol{c}} _{\boldsymbol{a}}, \hat{\boldsymbol{c}} _{\boldsymbol{m}}, \hat{\boldsymbol{z}} _{\mathbf{0}}^{\boldsymbol{t}}\right)= \operatorname{Softmax}\left(\frac{Q _{a u x} K _{main }}{\sqrt{d}}\right) \cdot V _{\text {prior }}
> $$
>
> Here, the definitions of Q, K, and V are given in Line 283 of the original manuscript. Notably, $\hat{\boldsymbol{z}} _{\mathbf{0}}^{\boldsymbol{t}}$ is an equivalent transformation of the diffusion network output prior $\boldsymbol{\epsilon} _{\boldsymbol{\theta}}^{\boldsymbol{t}}$ via Eq. (3) (Line 163 of the original text), which contributes to training stability. The LBA is designed to avoid matching errors caused by generative prior hallucination. Specifically, it first models the context between the original cross-view representations ($Q _{aux} , K _{main}$), and then calibrates the generative representation $V _{\text{prior}}$ using the learned relations, thus ensuring cross-view consistency.
>
> **The authors are encouraged to report the computational cost**
> We have added experiments on the number of parameters and FLOPs. Due to space limitations, please refer to Table 3 in https://anonymous.4open.science/r/anonymous-git-EEA2/README.md or our response to Reviewer Gwkd for detailed data. The coding and decoding complexity of SDiD surpasses existing diffusion-based baselines, and its encoding complexity is also competitive compared with non-generative methods. It should be acknowledged that the number of parameters and computational cost of the SDiD encoder have not yet reached the state-of-the-art level. Further lightweight optimization and deployment on edge devices will be key technical directions discussed in depth in future work.

---

> > ### Author Rebuttal · Reviewer_Debk · 2026-04-02
> >
> > I give "weak accept".

---

> > > ### Author Response · Authors · 2026-04-02
> > >
> > > Thank you for your recognition! We have added theoretical supplements for the ANE module in our response to Reviewer X1TH, proving that ANE is equivalent to minimizing the Wasserstein Distance. You are welcome to check it if you are interested.

---

### Decision · Program_Chairs · 2026-04-30

**Decision:**

Accept (regular)

**Comment:**

This paper proposes SDiD, a distributed stereo image compression framework based on shared diffusion priors, which addresses two critical challenges in stereo compression: cross-view inconsistency leading to downstream task degradation and prohibitive denoising costs of diffusion-based methods. All four reviewers assigned a score of 4 (Weak Accept), and the authors submitted a comprehensive, data-rich rebuttal that addressed nearly all raised concerns, including adding extensive new experiments on cross-resolution robustness, computational complexity, and comparisons with missing state-of-the-art baselines, as well as providing detailed mathematical derivations, module architectural clarifications, and theoretical analysis of the ANE module. The work is widely recognized for its clear motivation targeting asymmetric distributed coding scenarios, innovative shared diffusion prior paradigm that enables ultra-fast two-step sampling, strong empirical performance at ultra-low bitrates, and significant improvements in both perceptual quality and downstream depth estimation accuracy over existing methods. While there remain minor unresolved concerns—primarily regarding the completeness of theoretical guarantees for DPA training stability and ANE error propagation, as well as presentation issues that can be easily corrected in the final version—the core technical soundness and practical value of the contribution are undisputed. Based on the reviewer consensus and the thorough resolution of all major issues, I recommend Weak Accept.